# Phylodynamic of SARS-CoV-2 during the second wave of COVID-19 in Peru

Santiago Justo Arevalo ⓘ[1,2,3] ✉, Carmen Sofia Uribe Calampa[1], Cinthy Jimenez Silva ⓘ[4], Mauro Quiñones Aguilar[1], Remco Bouckaert[5] & Joao Renato Rebello Pinho ⓘ[2,6]

At over 0.6% of the population, Peru has one of the highest SARS-CoV-2 mortality rate in the world. Much effort to sequence genomes has been done in this country since mid-2020. However, an adequate analysis of the dynamics of the variants of concern and interest (VOCIs) is missing. We investigated the dynamics of the COVID-19 pandemic in Peru with a focus on the second wave, which had the greatest case fatality rate. The second wave in Peru was dominated by Lambda and Gamma. Analysis of the origin of Lambda shows that it most likely emerged in Peru before the second wave (June–November, 2020). After its emergence it reached Argentina and Chile from Peru where it was locally transmitted. During the second wave in Peru, we identify the coexistence of two Lambda and three Gamma sublineages. Lambda sublineages emerged in the center of Peru whereas the Gamma sublineages more likely originated in the north-east and mid-east. Importantly, it is observed that the center of Peru played a prominent role in transmitting SARS-CoV-2 to other regions within Peru.

More than 2 years have passed since the first cases of unexplained viral pneumonia were reported in the city of Wuhan. On March 11th, 2020, the World Health Organization (WHO) officially declared SARS-CoV-2 as a pandemic. On April 12th, 2023, there were 762 million confirmed cases worldwide[1].

The first confirmed case of COVID-19 in Peru was on March 6th, 2020, in a 25-year-old man coming from Europe. After this, three waves were reported in Peru (during the writing of this manuscript the fourth and fifth waves hit Peru). The first wave ranged from April, 2020, to November, 2020. A devastating second wave from January to June, 2021, caused >980,485 reported cases and 98,837 confirmed deaths[1]. Then, a third wave hit Peru causing the highest number of reported cases per day (up to ~50,000 cases per day), but the number of deaths were much less compared to the first and second waves. By November 30th, 2022, Peru had around 4.23 million of COVID-19 reported cases with 217,000 confirmed deaths[1].

Peru is among the countries most affected by the pandemic. In the first 90 days of the pandemic, the highest accumulated incidence rate in Latin America and the Caribbean was observed in Peru (5 426.3 cases/million), followed by Chile, Panama, Ecuador, and Brazil[2]. As of November, 2022, Peru still has the highest mortality rate in the world (6590 deaths/million)[1].

During the COVID-19 pandemic, several lineages appeared worldwide, but just some of them reached high global prevalence. The most important lineages, due to their relative frequencies and their apparently improved capacity of transmission[3–12] were designated as Variants of Concern (VOCs) or Variants of Interest (VOIs) by the World Health Organization[13]. Five SARS-CoV-2 variants were classified as VOCs (Alpha, Beta, Gamma, Delta, and Omicron) and some others as VOIs, such as Lambda and Mu that, together with the Gamma VOC, were the most prevalent in South America according to Nextstrain[14,15].

[1]Facultad de Ciencias Biológicas, Universidad Ricardo Palma, Lima, Peru. [2]Laboratório Clínico do Hospital Israelita Albert Einstein, São Paulo, Brasil. [3]Departamento de Bioquímica, Instituto de Química, Universidade de São Paulo, São Paulo, Brasil. [4]School of Biological Sciences, University of Auckland, Auckland, New Zealand. [5]School of Computer Science, University of Auckland, Auckland, New Zealand. [6]LIM03/07, Department of Gastroenterology and Pathology, University of São Paulo School of Medicine, São Paulo, Brazil. ✉e-mail: santiago.justo@urp.edu.pe

Since the second wave of COVID-19 cases in Peru, there has been a significant effort to sequence genomes[16,17], resulting in the reporting of ~300 genomes per week[17]. Thanks to this, the presence of six of the VOCs or VOIs (Alpha, Gamma, Delta, Lambda, Mu, and Omicron) were reported at the time in Peru[16–18]. From the VOIs and VOCs reported in Peru, Lambda was of particular interest due to its predominance during the deadliest second wave in Peru and because it was hypothesized that this VOI emerged in Peru.

Previous estimates of the origin of Lambda lineage put its MRCA between September and November, 2020[18], or around July 12th, 2020[7]. However, the location of origin for the Lambda lineage, as well as the date of its emergence, need to be carefully reevaluated considering the presence of Lambda genomes in countries beyond Peru.

Here, we investigated the dynamics of the COVID-19 pandemic in Peru with focus in the second wave. We use epidemiological data and Bayesian phylogenetics to estimate the date and country of origin of VOCI Lambda and to reveal the dynamics of the VOCIs Lambda and Gamma that dominated the second wave of COVID-19 in Peru.

## Results

### Three waves and multiple lineage replacements characterizes the COVID-19 pandemic in Peru

Until June 2022, the COVID-19 pandemic in Peru was characterized by three waves (Fig. 1a): (i) The first wave began in April, 2020, and stimulated the implementation of several lockdown measures in Peru, as shown by the high stringency index (SI > 90) reported in the COVID-19 Government Response Tracker (Fig. 1b)[19]. These measures were progressively diminished up to July, 2020, when a rise in the number of cases again encouraged the reimplementation of measures (SI > 80) (Fig. 1a, b). After the lowering of cases and the end of the first wave of COVID-19 in October, 2020, the stringency index continued to decrease until December, 2020, when it reached its lowest value since the beginning of the pandemic (SI < 60) (Fig. 1a, b). (ii) In January, 2021, a new rise of cases marked the beginning of the second wave and encouraged measures to be restarted (SI > 80) (Fig. 1a, b). Again, these measures were progressively diminished until the end of the second wave in July 2021 (SI < 70) (Fig. 1a, b). During the second wave, in February, 2021, the vaccination program began in Peru probably contributing to the decrease of cases and the end of the second wave (Fig. 1a, b), although the precise contribution of vaccinations needs to be investigated further. Importantly, the first two waves had a case fatality rate of ~10% (number of deaths divided by the number of cases) (Fig. 1a). (iii) In December, 2021, a third wave of COVID-19 cases hit Peru. But this time, just a modest increase of the stringency index was observed (SI around 70) (Fig. 1a, b). During this third wave, that spans up to March, 2022, the number of cases per day was much higher than in the previous waves (Fig. 1a). In contrast, the case fatality rate decreased to ~0.5%, meaning a reduction of ~95% deaths compared to the first and second wave (Fig. 1a). This is probably a result of the vaccination program that at the beginning of the third wave had around 30 million total vaccinations with ~70% of people with at least one doses (Fig. 1b)[20]. Consistently, effects of the vaccinations have been reported in other regions, showing between 46% to 98% reduction in deaths[21,22].

Five variants were classified as VOCs by the WHO (Alpha, Beta, Gamma, Delta, Omicron) and some others, including Lambda and Mu, as VOIs[13]. All of the mentioned VOCIs, except Beta, have been reported in Peru (Fig. 1c). From them, Lambda, Gamma, Delta, and Omicron reached prevalence >10% in at least 1 month (Fig. 1c). In contrast, Alpha and Mu were not so successful in Peru (Fig. 1c). During the first wave none of these variants had a high relative prevalence. On the other hand, the second wave was marked by the prevalence of Gamma and Lambda that replaced the pre-existing lineages (Fig. 1c). At the end of the second wave, Delta replaced both Gamma and Lambda and had the highest prevalence until the beginning of the third wave when

Omicron replaced Delta, marking the third lineage replacement in Peru (Fig. 1c).

### The origin of Lambda and its initial expansion in South America

Gamma and Lambda dominated the second wave in Peru (Fig. 1). However, although the origin of Gamma and its related lineages has been extensively studied[4,23–26], the origin of Lambda has received less attention. It has been hypothesized that Lambda has emerged in Peru[18]. However, this hypothesis has been supported only by the fact that most of the first genomes belonging to this variant were isolated in Peru (Fig. S1).

To analyse more in-depth the most likely country of origin of Lambda, we first identified the candidate countries where this variant could emerge. Based on evidence that Lambda has a comparable ability to evade the immune response with respect to other contemporaneous VOCIs (i.e: Alpha, Gamma)[7,27], we defined two criteria for identifying countries as potential origins of Lambda: (i) Lambda must have been sampled in at least two cities of the country of origin before April, 2021 and (ii) in these cities at least 15% of sequenced cases sampled before April, 2021 must have been classified as Lambda (that was the time when in Peru the Lambda variant had reached >50% prevalence, see Fig. 1c). Based on these two considerations, we identified six countries where Lambda could have originated (Figs. S2 and S3). Five of the six countries are in South America, giving this region the most likely region where Lambda originated. It is important to note that countries and cities in South America could be undersampled, with some of them without available genomes to include in the analysis. Thus, we cannot rule out these countries and cities as possible locations of origin of Lambda.

To determine on which of these six countries was most probable that Lambda has emerged, we first calculated the relative prevalence of Lambda by week in these countries and adjusted the prevalence by local polynomial regression (LOESS[28]) (Fig. 2a). Then, using the adjusted prevalence we estimated the number of Lambda cases by week in each country (Fig. 2b). This analysis showed that between November and December, 2020, three countries (Argentina, Chile and Peru) reached at least 1% prevalence of Lambda (Fig. 2a) and thus unreported Lambda cases could have already been circulating (Fig. 2b). On the other hand, the other three countries (Mexico, Ecuador and Colombia) reached a Lambda prevalence of 1% after February, 2021 (Fig. 2a). It is unlikely that Lambda prevalence in Mexico, Colombia, and Ecuador were underestimated because the number of sequenced genomes and sampling proportion were relatively similar to those in Chile, Peru and Argentina between October, 2020, and April, 2021 (Fig. S4).

The above analysis showed that Peru, Chile, and Argentina could be the location of origin of the Lambda variant. Thus, to determine the most likely location of origin, we conducted phylogeographic analyses. First, to reduce the sampling bias in the dataset of Lambda genomes we randomly take one Lambda genome per each ~7400 Lambda cases per week of each country (this procedure was repeated three times to yield three different samples). This sampling procedure reduces the computational effort required and, most importantly, allow us to improve the correlation between number of cases and number of genomes in the dataset ($r^2 = 0.97$ vs $r^2 = 0.12$) (Fig. S5). However, it is important to recognize that variable percentage of true cases recorded in each country can also introduce another layer of bias.

Then, we analysed the percentage of resolved quartets (groups of four sequences randomly extracted from the alignment with at least one of the three possible fully resolved tree topologies with a likelihood distinguishable from the other two) by likelihood mapping[29] and the root-to-tip distance vs. sequence sampling time correlation of datasets of Lambda genomes in increasing months from January, 2021 to October, 2021 (Fig. S6). Correlations between root-to-tip genetic

distance vs sampling time showed that samples that covered genomes from January, 2021 to September, 2021 presented a correlation coefficient (R-value) >0.55 (Fig. S6). Additionally, likelihood mapping showed that these samples (from January, 2021, to September, 2021) have >54% resolved quartets, while not optimal, it can be considered appropriate to perform phylogenetic analysis (Fig. S6). In summary, three samples with one genome per each ~7400 Lambda cases covering January, 2021, to September, 2021, from the three countries

considered were used to determine the most likely country of origin of the Lambda variant.

The phylogeographic analysis of the three Lambda samples showed consistent estimations of the date of origin (root date) (Fig. 3a) and substitutions rates in the ranges extensively reported for SARS-CoV-2 (Fig. S7). Considering all three samples, the overall 95% highest posterior density (HPD) of the root date puts the origin of Lambda between May and October, 2020, indicating that Lambda had already

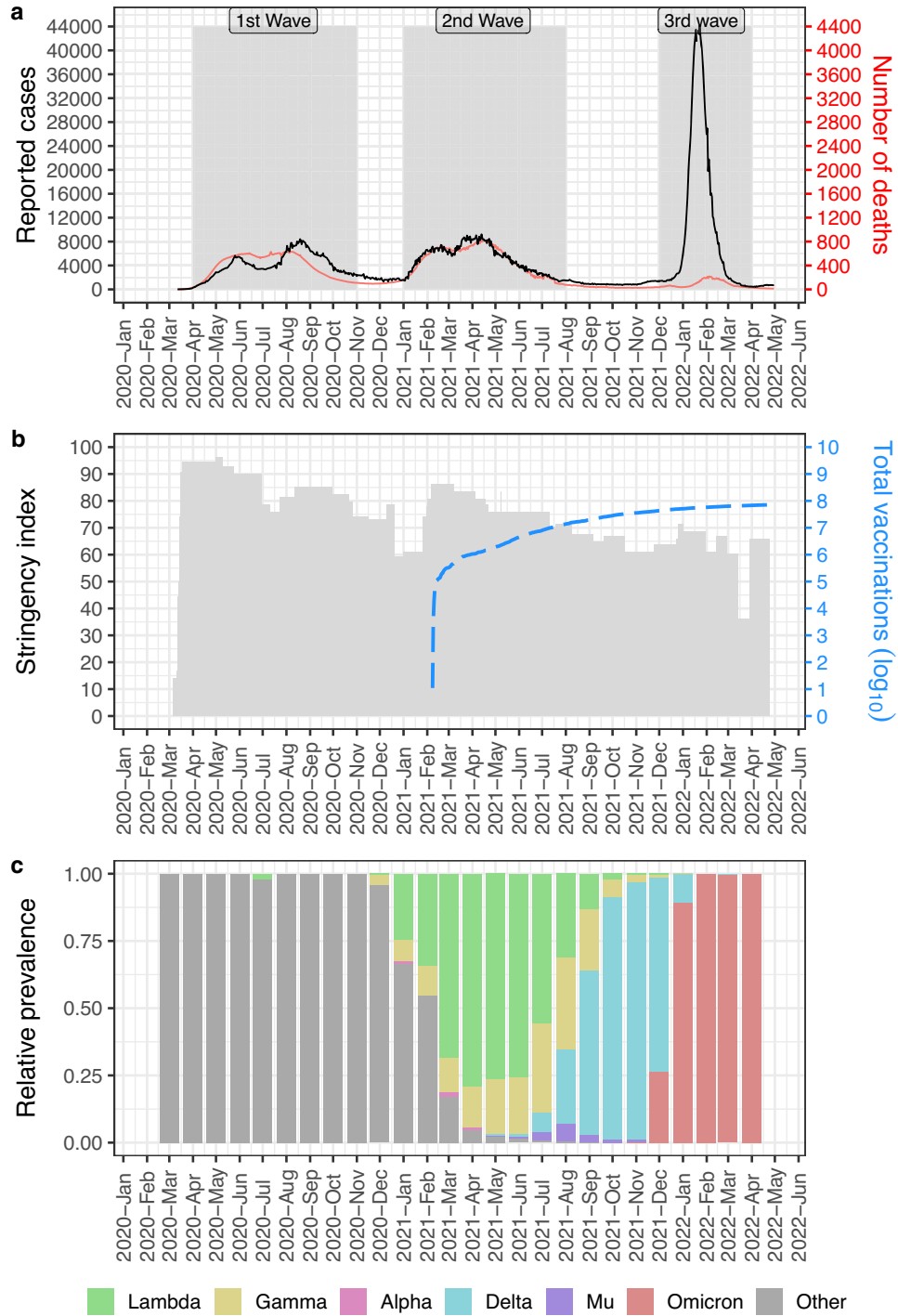

**Fig. 1 | COVID-19 pandemic in Peru was characterized by three waves and lineage replacements. a** The 14-day average of the number of cases or deaths are shown in black and red, respectively. Three waves of COVID-19 cases can be observed. **b** Bars in gray represent the stringency index level (Hale et al. 2021) and

the blue discontinuous lines represent the log of the total vaccinations. **c** Relative prevalence of VOCIs in time showing episodes of lineage replacements during the COVID-19 pandemic in Peru.

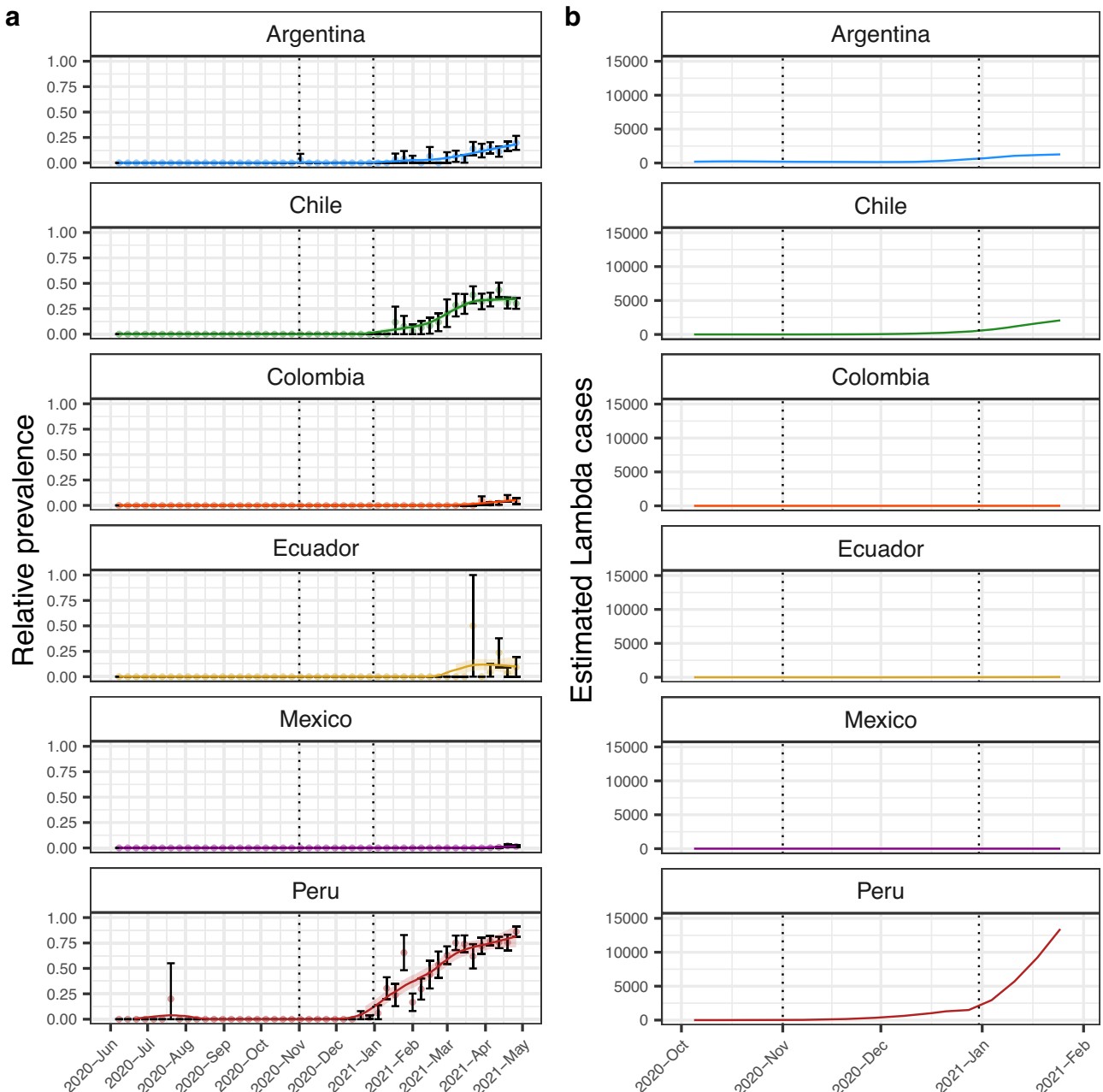

**Fig. 2 | Lambda VOI was already circulating in Argentina, Chile and Peru between November and December 2020. a** Weekly relative prevalence of Lambda in six different countries. Each point represents the calculated relative prevalence of Lambda by week with 95% confidence intervals as error bars. Relative prevalence adjusted by local polynomial regression is shown as lines in each graphic. **b** Estimated Lambda cases in the time, estimations were obtained by multiplication of the 14-day average number of cases by week with the adjusted relative prevalence in **a**. Between November and December, 2020, In Argentina, Chile and Peru the estimated Lambda cases is >0. Dashed vertical lines in **a** and **b** mark the beginning of November, 2020, and the end of December, 2020, to improve visualization.

been circulating before the second wave began in Peru and even during the first wave (Figs. 1a and 3a). Besides this, the ancestral state reconstruction of the root showed Peru as the most likely country of origin of Lambda (>50% taking into consideration the three samples), followed by Argentina (Fig. 3b).

Additionally, we noted that during the sampling procedure some available genomes from the weeks before Lambda reached 7400 estimated Lambda cases were not considered (Fig. S8). These genomes could be informative about the origin of Lambda. To analyze the effect of including these genomes in the sample, we took six additional samples where we add one available genome per week from the weeks before Lambda reached 7400 estimated Lambda cases (Fig. S9) and

performed phylogeographic analyses with these samples. Although the inclusion of these genomes affected the correlation between the number of cases and the number of genomes during the first weeks, the conclusion that the Lambda variant likely originated in Peru and the estimated dates of its origin were almost unaffected (Fig. S10). It is worth nothing that for one sample (sample 3 in Fig. S10), the probabilities of being the origin for the three countries were similar, indicating that this sample did not contain enough information to differentiate between the three possible location of Lambda variant's origin.

We then further explored the Lambda migration patterns between Peru, Argentina, and Chile, assuming that either Peru or Argentina was

the place of origin. Trees from the posterior distribution with Peru as the root indicated that between early August and early December, 2020, and between early September and mid-December, 2020, (95% HPD intervals of the first transition between states), Lambda reached Argentina and Chile from Peru, respectively (Fig. 4a). On the other hand, trees where Argentina was the root differed in that Lambda reached Peru from Argentina between late July and mid-November, 2020. But, similarly to the previous hypothesis, Lambda reached Chile from Peru between September and December, 2020 (Fig. 4b). In both scenarios, most Lambda genomes from Argentina or Chile are clustered in a single monophyletic group indicating that propagation of Lambda in both countries was mainly due to local transmission (Fig. 4). Lambda, together with Gamma, governed the second wave in the three countries, reaching relative prevalence of around 25% in Argentina and Chile, and around 90% in Peru (Fig. 2).

Taken together, our phylogeographic analysis showed that the most likely hypothesis of Lambda origin is that it emerged in Peru between May and October, 2020. Then, from Peru, it reached Argentina and Chile. Once in these countries, Lambda propagates inside these countries mainly by local transmission contributing with several COVID-19 cases during their respective second waves.

## Different Lambda and Gamma sublineages circulated in Peru

Lambda, together with Gamma, dominated the COVID-19 pandemic in Peru, before being replaced by the Delta VOC (Fig. 1c). Whereas Lambda reached relative prevalence around 90% representing >40,000 weekly cases, Gamma reached maximum prevalence around 25% that represented around 5000 weekly cases (Fig. 5a, b, insets).

We began the analysis by determining if we could identify sublineages of Lambda and Gamma. A maximum likelihood phylogeny of all the available Peruvian Lambda genomes shown two well-supported sublineages (here named SubL1 and SubL2) (Fig. 5a). Almost all the SubL2 genomes presented a T in position 28849, whereas most of SubL1 have a C in this position (Fig. S11). Both were reported in all the six Peruvian regions that we considered in this manuscript (hereafter we refer to six Peruvian regions: south, center, north, south-east, mid-east, north-east. Each of these regions groups one or more Peruvian states, see Table S1 for information about which states are grouped in each region) (Fig. 5a). When we analysed the relative prevalence adjusted by LOESS of each of these sublineages by region, we observed that both (SubL1 and SubL2) followed similar patterns in all the regions reaching the peak of prevalence between March and July, 2021 (Fig. 5c). In all six regions the prevalence of SubL2 was slightly less than SubL1 (Fig. 5c). Interestingly, the sum of the prevalence of the Lambda sublineages in five of the six regions was much >50%; however, in the north-east this sum did not reach 50% indicating that, opposite to all the other regions, Lambda was not the most prevalent lineage during the second wave in the north-east of Peru.

In the case of Gamma, the maximum likelihood tree of all the Peruvian Gamma genomes shown three sublineages (SubG1, SubG2, and SubG3), each of them had at least one reported genome from each Peruvian region indicating that all the sublineages circulated in all the Peruvian regions (Fig. 5b). The first Gamma genomes reported in Peru belong to the sublineage SubG2. This sublineage has a combination of T, C, C, A, T, C in position 3049, 10,116, 22,298, 23,599, 23,604, and 25,613, respectively (Fig. S12). SubG2 caused a peak with most of the Gamma cases in the center, north, north-east and south-east of Peru (Fig. 5d). This peak was slightly before in the north-east than in the other three regions, probably indicating that SubG2 reached the north-east before the other regions (see below the phylodynamic analyses) (Fig. 5d). Of note, on the mid-east, SubG2 did not cause most of the Gamma cases, instead the sublineage SubG3 was responsible for this (Fig. 5d). SubG3 has three conserved nucleotide substitutions in respect to SubG2, positions 10116, 22298, and 23604 are T, A, and C, respectively (Fig. S12). Furthermore, in all the regions except in the

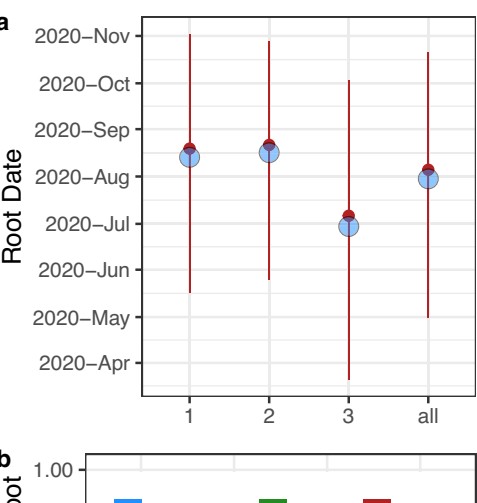

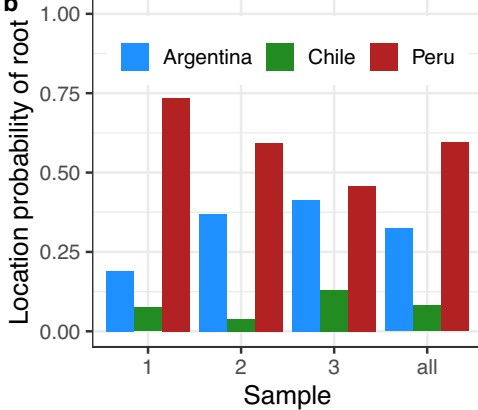

**Fig. 3 | Lambda most likely emerged between May and October, 2020, in Peru.** **a** Lines represent the high posterior density 95% of the root date which represents the time of origin of the lineage Lambda. Blue circles and red points represent the mean and the median of the posterior distribution, respectively. **b** Location probability of the root that represents the MRCA of the Lambda lineage. In **a** and **b**, the results of three analysis with different samples are shown (1, 2, and 3) together with the combined results of these three samples (all). The number of trees (*n*) from the posterior distributions that were analysed were 391, 349, and 366 for samples 1, 2, and 3, respectively.

mid-east SubG3 and SubG1 followed similar patterns (in the south, the three Gamma sublineages followed similar patterns) (Fig. 5d). In the center and north-east, SubG3 and SubG1 caused considerable peaks of Gamma cases (Fig. 5d). Opposite to SubG3, SubG1 has positions 10,116, 22,298, and 23,604 equals to SubG2 but positions 3049, 23599 and 25613 are A, G, and T, respectively (Fig. S12).

## Lineages emerged in the center, north-east and mid-east

To determine the most likely Peruvian region where Gamma and Lambda sublineages emerged, we performed Bayesian phylogeographic analyses with Peruvian regions as discrete states. We first performed and undersampling procedure, similar to the analysis of the country of origin of Lambda (see above), taking into consideration the number of cases in each region to reduce sampling bias (Figs. S13 and S14). The undersampling was done by triplicates. Also, we analysed the root-to-tip distance vs. sequence sampling time correlation and the percentage of resolved quartets of these samples (Table S2 and S3). Sampling based on the number of cases allowed us to obtain a high correlation between the number of cases and the number of genomes (*R* > 0.70). However, despite the sampling procedure improving the correlation, a low number of Gamma sublineage genomes in some weeks with a relatively high number of cases prevented the correlations from reaching even higher *R*-values (Fig. S14). Additionally, the

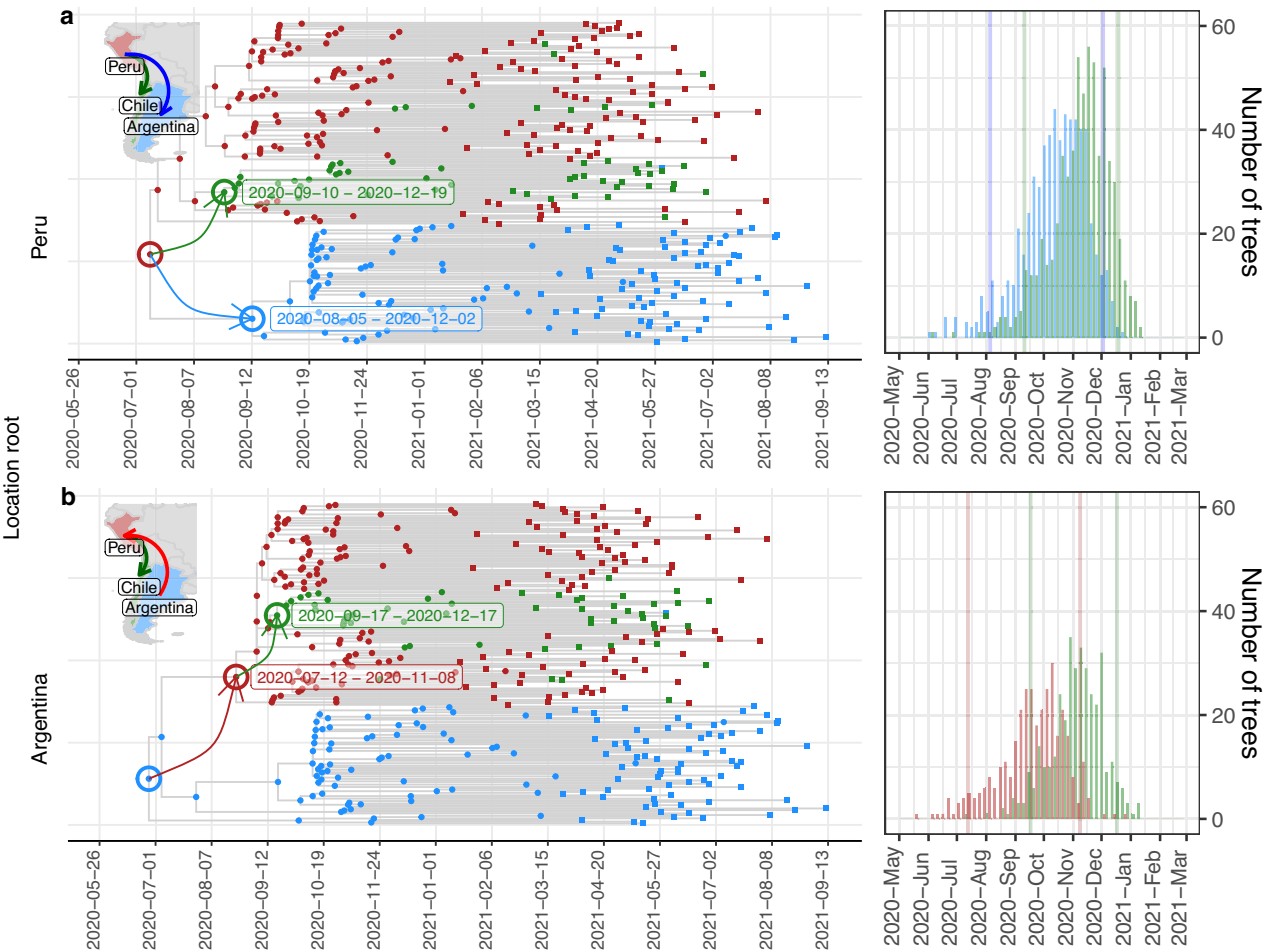

**Fig. 4 | Hypotheses of the origin of Lambda and its initial expansion in South America.** Maximum clade credibility tree from trees of the posterior distribution where Peru (**a**) or Argentina (**b**) was estimated as the root. Labels indicated the high posterior density 95% of the dates of the first node of each country that indicate the estimation of the dates of origin of Lambda in each of these countries. Nodes are colored according to the country (Peru = Red, Argentina = Blue, Chile = Green). Arrows indicate from where to where the first transitions to each country occurred. At the left as inset, a graphic map summarizing the transitions of Lambda between countries is shown. At the right, distribution of the date of the node representing the first transition between Peru to Chile (green), Peru to Argentina (blue) or Argentina to Peru (red) is displayed. Vertical lines indicate 95% high posterior density. The number of trees analysed (*n*) from the posterior distributions was 636 when Peru was the root, and 341 when Argentina was the root.

root-to-tip vs. sequence sampling time showed correlation coefficients (*R*-values) >0.5 (Table S1). However, the percentage of resolved quartets was very low in the sample of SubL1 (23%), whereas other sublineages presented at least 44% resolved quartets (Table S2). Because of this and because position 28849 is only partially conserved in SubL1, we decided to perform a discrete phylogeographic analysis of each Gamma sublineage and Lambda, joining SubL1 and SubL2 (named SubL1 + L2).

These analyses showed that the MRCA of SubL1 + L2 most likely existed in the center between September and November, 2020 (Fig. 6a, b). The origin of the Gamma sublineages, SubG1 and SubG3, was confidently traced to the north-east and mid-east, respectively, with their dates between March and April, 2021, and December and February, 2021, respectively (Fig. 6a, b). In the case of SubG2, estimations of the MRCA were more uncertain with its date ranging from June to November, 2020, and its location with similar probabilities between the center and the north-east (both ~37.5%) (Fig. 6a, b). The results from the three samples were consistent, except for the third sample of SubG2, which showed similar probabilities for the mid-east and north-east as the origin (Fig. S15). It remains and open question whether the Gamma sublineages emerged in Peru or were introduced from another country such as Brazil.

## The center was the main exporter of lineages to other Peruvian regions during the second wave

After determining the most likely origins of the sublineages, our next step was to better understand the dynamics between Peruvian regions. We performed these analyses on three different samples, which produced consistent results (Fig. S16). These samples were combined, and the resulting distributions are presented in the following text.

Regarding SubL1 + L2, the center was the origin of most of the transitions between regions with a median of 65 transitions per tree (95% interval of transitions: 51–76) from a median of total transitions per tree of 73 (95% interval of transitions: 62–84) (Fig. 6c). The north represented the destination of most of transitions where the center was the origin with a median of 26 transitions per tree (95% interval of transitions: 18–35) followed by the south-east, south and mid-east (12–21, 9–16, and 3–11, 95% interval of transitions per tree, respectively) (Fig. 6c). These results suggest that the center was the main exporter of the Lambda lineage to other Peruvian regions.

In the case of SubG1, the median of total transitions per tree was 33 (95% interval transitions: 24–55) (Fig. 6c). As shown in Fig. 6a, the SubG1 most likely emerged in the north-east of Peru. Most of the transitions where north-east was the origin had the center as the destination with a median of 11 transitions per tree (95% interval

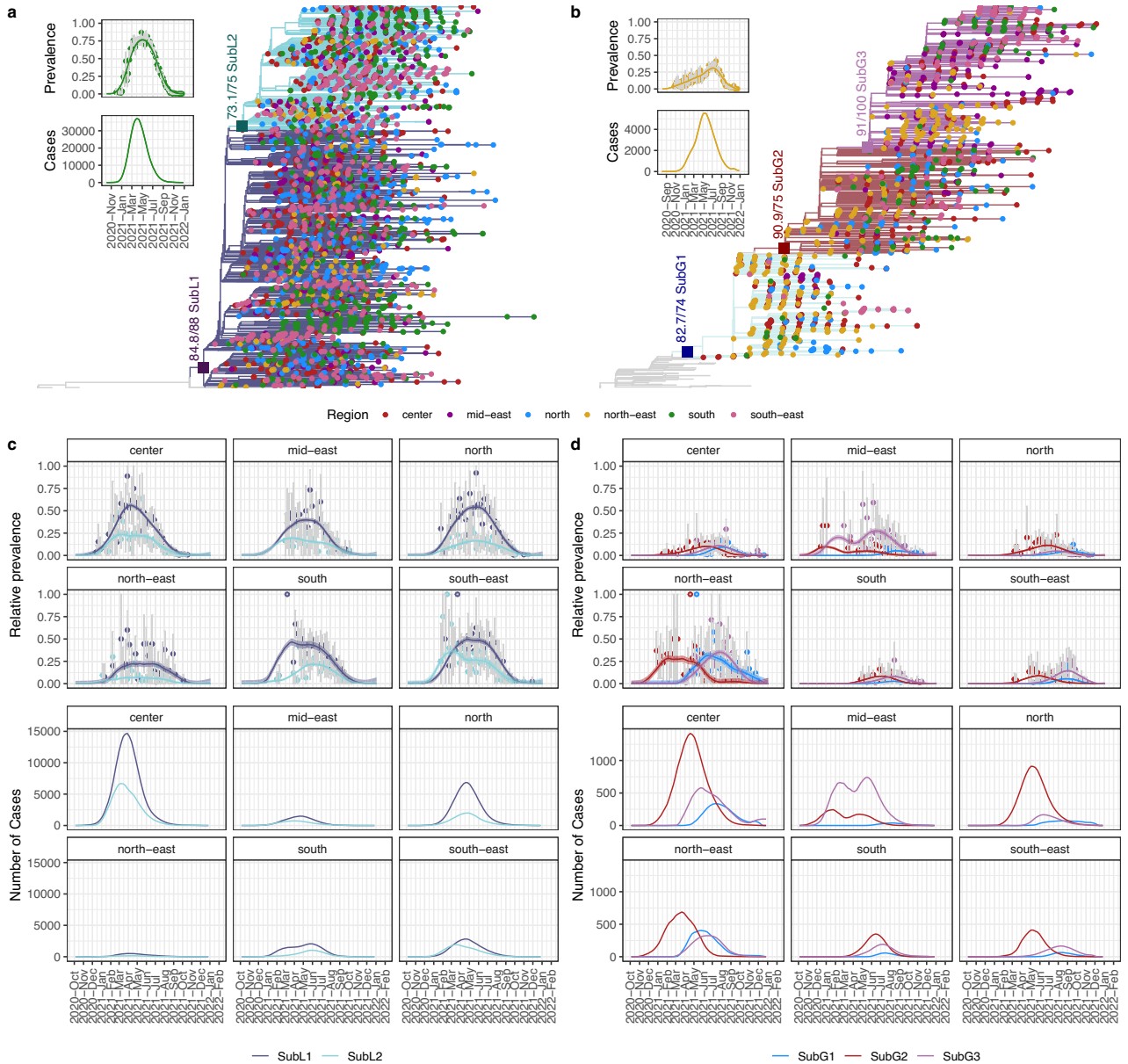

**Fig. 5 | Lambda and Gamma sublineages circulated during the second wave of COVID-19 in Peru.** Maximum likelihood trees of Lambda ($n = 3461$) (**a**) or Gamma ($n = 1674$) (**b**) genomes from Peru showing two (Lambda) and three (Gamma) clades with high support (SH-aLRT/bootstrap) representing sublineages. Tips points are colored according to the region where the genome was collected. At the left, insets are showing the overall prevalence and estimated cases of Lambda or Gamma by week in Peru. Error bars in relative prevalence indicate the 95% confidence interval. **c**, **d** are showing the weekly relative prevalence (above) and estimated number of cases (below) of each sublineage in each peruvian region. In the relative prevalence graphics, points represent the calculated relative prevalence by week and the error bars represent the 95% confidence intervals, lines represent the adjusted relative prevalence by local polynomial regression (LOESS).

transitions: 5–26) (Fig. 6c). Furthermore, the center was the likely origin of other transitions of SubG1 to the other regions, especially to the north and the south, with a median of 7 and 3 transitions per tree to the north and south, respectively (2–11 and 1–3, 95% interval transitions per tree, respectively) (Fig. 6c). Thus, it was likely that after the transition of SubG1 from north-east to the center, SubG1 reached the other regions from the center.

As previously mentioned, SubG2 could have emerged in the center, north-east or even in the mid-east (Fig. 6a, S15). This sublineage had a median of 56 transitions per tree (95% interval of transitions: 45–71) (Fig. 6c). The center and the north-east had the highest concentration of estimated transitions for this sublineage, with the center and north-east being the origin of a median of 21 and 25 transitions per

tree, respectively (7–36 and 5–49, 95% interval transitions per tree, respectively) (Fig. 6c). Interestingly, transitions from the north-east to the center were estimated to be between 4 and 26 transitions per tree raising the possibility that if the north-east was where SubG2 emerged, then it moves at least four times to the center and from here SubG2 could reached the other Peruvian regions, mainly the north (95% interval of transitions: 1–13) (Fig. 6c).

Finally, SubG3 exhibited a median of 53 transitions per tree (95% interval transitions: 39–67) (Fig. 6c). In contrast to the other sublineages, where the center played a central role as an exporter, SubG3 had the mid-east as its main source region, with a median of 26 transitions per tree (95% interval transitions: 15–39) (Fig. 6c). SubG3, which likely originated in the mid-east was mainly transferred to the

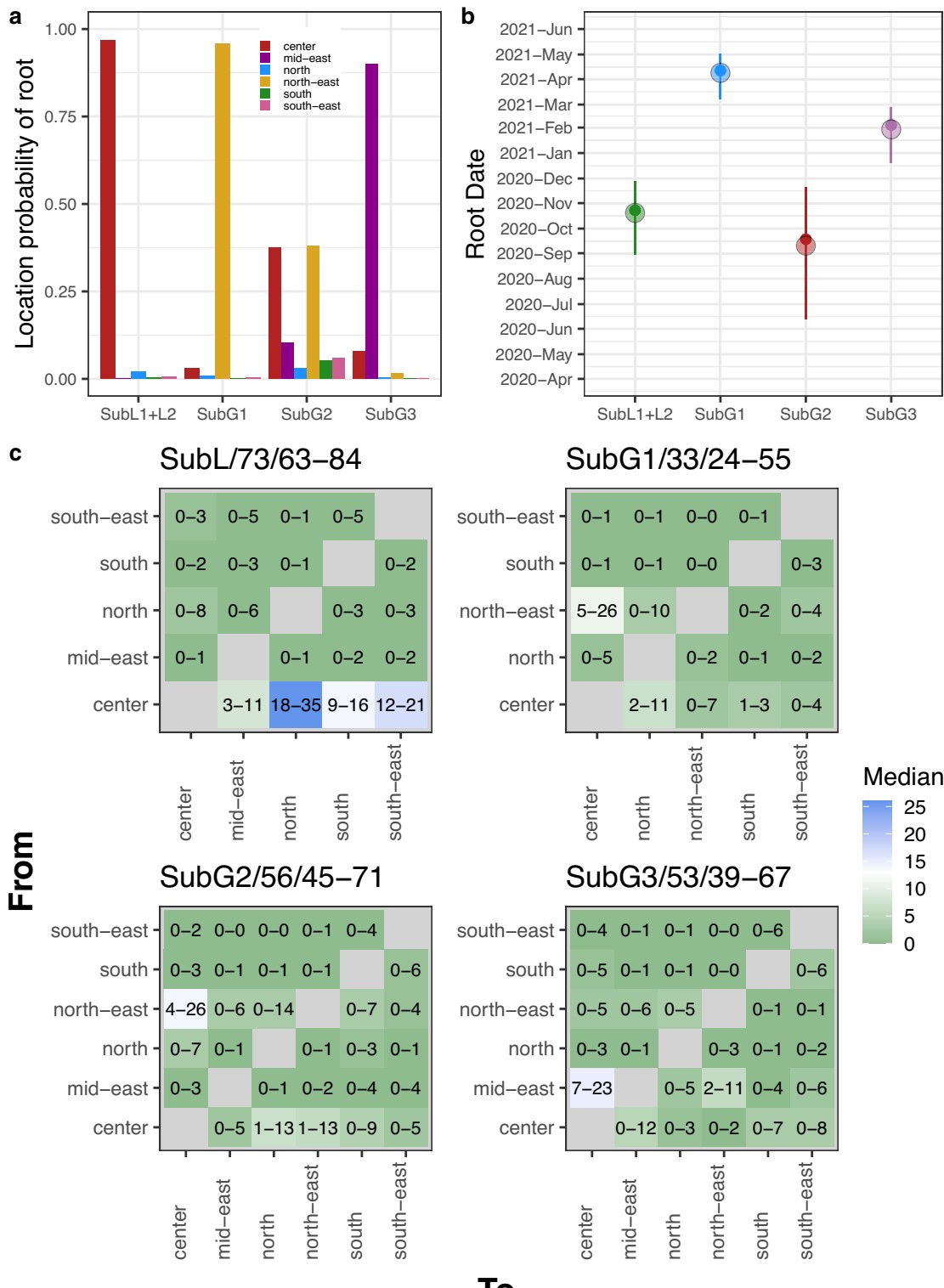

**Fig. 6 | Phylodynamics of Lambda and Gamma sublineages between Peruvian regions.** **a** Location probability of the root that represents the MRCA of the sublineage. Bars are colored according to the Peruvian region. **b** Lines represent the high posterior density 95% of the root date which represent the time of origin of the sublineage. Transparent circles and small filled points represent the mean and the median of the posterior distribution, respectively. **c** Matrices of the 2.5% to 97.5% fraction of the distribution of transitions per tree between Peruvian regions. In the *y*-axis is depicted the region of origin of the transition and the *x*-axis showed the region of destination. Squares are colored according to the median of the distribution of transitions per tree. Titles of each panel shown the name of the sublineage, the median of the distributions of transitions per tree between regions and the overall 2.5% to 97.5% fraction of the distribution of transitions per tree. Each separated by "/". The number of trees (*n*) from the posterior distributions that were analysed were 400 for each of three samples that were merged to form the final posterior distribution showed in this figure.

center, with a median of 15 transitions per tree (95% interval of transitions: 8–23) (Fig. 6c). The next most common transition was from the mid-east to the north-east, with a median of 6 transitions per tree (95% interval of transitions: 2–11).

## Discussion

In this study, we investigated the origin of the Lambda lineage and the dynamic of the COVID-19 pandemic during the second wave in Peru. We showed that Lambda was most likely originated in Peru before the second wave. After its origin, from Peru it reached Argentina and Chile where local transmission contributed to a raise of COVID-19 cases in these two countries. When we analysed more in depth the second wave in Peru, we determined that at least two and three sublineages of Lambda and Gamma, respectively, circulated in Peru. All of these sublineages were reported in all the regions of Peru. Furthermore, our analyses suggest that Lambda sublineages emerged in the center of Peru, whereas two and one Gamma sublineage emerged in the north-east and in the mid-east of Peru, respectively. Finally, we showed that despite the diverse regions where sublineages could emerge, in most of the cases the center of Peru was the main exporter of SARS-CoV-2 to other Peruvian regions during the second wave.

Phylogeographic analyses of the Lambda origin showed Argentina as a second probable country of origin (Fig. 3b). In this hypothesis, Lambda reached Peru from Argentina between July and November, 2020 (HPD 95% July 12th–November 08th, 2020) (Fig. 4b). This scenario is inconsistent with a Lambda genome collected in Peru on July 21st, 2020, (EPI_ISL_5934936). Therefore, favouring the hypothesis that Lambda emerge in Peru and then move to Argentina and Chile (Fig. 4a). Additionally, our estimation of the date of origin of Lambda is consistent with previous estimations[7,18] and with the Lambda genome collected in July (Fig. 3a). However, the estimation of the MRCA of Lambda between September and November by Padilla-Rojas et al., 2021 is inconsistent with the existence of this genome. Nevertheless, due to the fact that the genome EPI_ISL_5934936 was submitted on November 5th, 2021 (15 months after its annotated collection date) we cannot rule out the possibility of an incorrectly recorded collection date for this genome.

Sampling bias in both discrete and continuous phylogeographic have been extensively studied[30–32]. Small number of genomes in countries where the variant of interest could have first emerged can obscure the real location of origin of this variant. In the worst scenario, the absence of genomes from those countries impedes formal inclusion in phylogenetic analysis (although if travel connections are known, they can be included[33]). In the best scenario, a carefully planned sampling approach will consider the number of cases in each region of interest to sampling each region proportionally to its number of cases. Thus, the sample used for the inference will reflect the real time and space distribution of the lineage[32]. However, this is difficult to achieve, and it is even more difficult when the analysis involves different countries. To overcome this, we used an undersampling approach similar to others described in the literature[34–36] to improve the correlation between the number of cases and number of genomes by region of interest.

It is also important to note that, even when maintaining the correlation between the number of cases and number of genomes, other biases may be present, such as variable percentage of true cases reported in different regions of interest. Additionally, the undersampling procedure can be challenging when there is a significant difference in the number of sequenced genomes between the regions of interest. For example, if we have two regions of interest with the same number of cases in a week, but one of them has ten times fewer genomes than the other, we must downsample the genomic information of one of the regions by ten times to maintain the correlation. Therefore, depending on the number of genomes, this can significantly impact the conclusions that can be extracted from the available genomes.

Phylogeography together with other sources of information such as migration rates, mean air traffic, travel information had been shown to be useful to give more robust results than genomic information alone[32,33,37]. However, for several countries, such as Peru, this information is not readily available.

Different efforts of sequencing exist not just at the global level. Even within country, different states are more capable to sequence genomes than others. In the case of Peru, in the original dataset we observed absence of correlation between number of estimated cases of the sublineages and the number of genomes of each sublineage. Thus, the same undersampling approach help us to increase the correlation that will reduce the sampling bias of the phylodynamic analyses.

During the second wave of COVID-19 in Peru Alpha, Gamma and Lambda were reported in Peru (Fig. 1c). Alpha never reached higher prevalences. Thus, different to other countries were Alpha surpassed the prevalence of pre-existing lineages[38–42], in Peru Alpha maintained low prevalence despite being reported in several regions (center, mid-east, north, and south-east). Alpha has been successful in several countries but not in all that it reached. For example, in Brazil it could not displace Gamma[43] and in Nigeria it was not as successful as Eta[44]. In this context, if Lambda and Eta had been reported in a timely manner in South America and Africa, respectively, they probably would have reached the category of Variant of Concern.

Most of the Peruvian regions were dominated by Lambda during the second wave (Fig. 5). However, in the north-east of Peru, Gamma was dominating over Lambda (Fig. 5). The higher prevalence of Gamma on the north-east of Peru and the fact that two of three Gamma sublineages that circulated in Peru likely emerged in the north-east (a third Gamma sublineage emerged in the mid-east, that is geographically near to the north-east) (Fig. 6a) raise the possibility that Gamma was successful in this region due to repeated introductions from the Brazilian border where Gamma dominated[23]. Additionally, it was shown that Gamma had a higher transmission rate in a population with high seroprevalence against other lineages[23,26] as it was the north-east of Peru[45,46]. A combination of these two facts could be responsible of the higher prevalence of Gamma in this region.

Our results showed that the center was the main exporter of lineages to other Peruvian regions during the second wave (Fig. 6c). Even sublineages that emerged first in regions different to the center were transported to this region and from this reached other Peruvian regions. This agrees with the fact that the center of Peru concentrates 30% of the total population in the country[47]. Thus, during the second wave the center reported most of the COVID-19 cases (Fig. 5c, d). Furthermore, during 2021 the center had 63% of both cargo and passenger transport[48].

At the final of the second wave, Delta was reported in Peru (Fig. 1c). Although it did not cause a large peak of cases and deaths, Delta replaced Lambda and Gamma and dominated the prevalence in Peru between the second and third waves. Delta was very successful in replacing pre-existing lineages in several countries[49,50] including Gamma in Brazil[43], Beta in South Africa[51], and Eta in Nigeria (and West Africa)[44]. It has been hypothesized that these replacements occurred due to its higher transmissibility and its ability to better evade the immune responses elicited by vaccination[52–54]. Supporting this, we can observe a steady increase in Delta´s prevalence despite the increasing number of vaccinations (Fig. 1).

## Methods
### Epidemiological dynamics
Analyses of COVID-19 cases, deaths, stringency index and vaccinations at the country level were done based on the information of "Our

World in Data"[19,20] (https://github.com/owid/covid-19-data/tree/master/public/data). The number of cases by Peruvian regions were obtained from the "Plataforma Nacional de Datos Abiertos" available in https://www.datosabiertos.gob.pe/dataset/casos-positivos-por-covid-19-ministerio-de-salud-minsa. For convenience, the geographical locations were aggregated as shown in table S3. The relative prevalence of VOCIs or sublineages was calculated based on data from GISAID[55] (www.gisaid.org). This data comprises 9 833 385 individual metadata of genome sequences with collection dates ranged from 2019-12-24 to 2022-04-30. The accession codes of the sequences and associated metadata for relative prevalence calculations are available in GISAID's EpiCoV database under EPI_SET_ID accession numbers EPI_SET_230526uh, supplementary data 1. Confidence intervals of the relative prevalence were calculated using the formula for a population proportion. The relative prevalence was smoothed using Local Polynomial regression[28] with the function "loess" in R, with a degree of 1, a span of 21 divided by the number of total weeks analysed, and other parameters set to default. The smoothed relative prevalence was used to estimate the number of reported cases belonging to each VOCI or sublineage.

### Assessment and selection of the genomic dataset

Lambda genomes from Peru, Chile and Argentina, and Gamma genomes from Peru were obtained from GISAID. This data comprises 9266 individual genome sequences and associated metadata. The accession codes of these sequences are available in GISAID's EpiCoV database under EPI_SET_ID accession numbers EPI_SET_230526dk, supplementary data 2. These sequences were aligned using ViralMSA.py[56,57] against the reference SARS-CoV-2 genome with GISAID code: EPI_ISL_406801 from nucleotide 203 to 29 674. After this, sequences with >290 Ns and/or >2% gaps were removed from the alignment. Then, the genomes were analyzed using Nextclade[14,15] and genomes classified as "bad" or "mediocre" were discarded. The genomes that passed the filter were used for subsequent analyses.

### Subsampling strategy to determine the origin of Lambda VOI

Since there was no correlation between the number of genomes and the number of reported cases, and given the large size of the Lambda dataset, a reliable and full Bayesian inference approach could not be suitable to determine the country of origin of Lambda. To address this issue, we took a series of samples with genomes collected from January, 2021 to April, May, June, July, August or September, 2021 based on the estimated number of Lambda cases by country per week. For each monthly interval, three samples consisting of ~200 genomes each were taken while maintaining the correlation between the number of genomes and the estimated number of Lambda cases. Correlations were estimated by Pearson coefficient. Additionally, to assess the impact of including genomes from weeks prior to when Lambda reached 7400 estimated cases, we took additional samples by adding one genome per week from those earlier weeks to the previously mentioned samples. Then, with each sample we constructed a maximum likelihood phylogeny using IQ-TREE2[58] with the substitution model GTR + F + I. These trees were used to evaluate the root-to-tip distance vs. sequence sampling time correlation using Tempest[59] and the same samples were used to perform likelihood mapping analyses[29] also implemented in IQ-TREE2 using 10000 quartets and the substitution model GTR + F + I.

### Bayesian phylogeographic analysis to determine the origin of Lambda

The time to the most recent common ancestor (MRCA) of each node and the pattern of SARS-CoV-2 spread in each of the samples from January to September, 2021 were estimated using the Bayesian discrete phylogeographic model[60]. We considered possible migrations between three demes (Peru, Chile and Argentina). We assumed that the transition rates between locations were reversible. For the analyses, we used the coalescent tree prior Bayesian integrated Coalescent Epoch PlotS (BICEPS)[61], the General Time Reversible (GTR) substitution model with 4 gamma categories and a strict clock[62] with a uniform substitution rate prior distribution between 1.0E-4 and 0.01 substitutions per site per year (s/s/y). We assumed one partition during the analyses.

### Phylogenetic analyses of Lambda and Gamma genomes from Peru

We analysed separately the Gamma and Lambda genomes from Peru. Each dataset (all the Lambda or Gamma genomes from Peru that passed the filters described in the section "Assessment and selection of the genomic dataset") was used to reconstruct a maximum likelihood phylogeny using IQ-TREE2 employing the GTR + F + I substitution model. Branch support was assessed using the ultrafast bootstrap approximation and the Shimodaira-Hasegawa-like procedure both with 1000 replicates. Sublineages were identified from these trees based in the following criteria: (i) ultrafast bootstrap approximation >70%, (ii) Shimodaira-Hasegawa support >70%, (iii) at least 400 genomes must be contained in these groups. To identify characteristic mutations in the sublineages, we searched for nucleotides that had >80% identity in one sublineage and <20% in the other sublineages.

### Subsampling strategy for the phylodynamic analysis of Lambda and sublineages of Gamma inside Peru

Similarly, to the analysis of the Lambda origin, the absence of correlation between number of genomes and the number of reported cases by Peruvian region increases the risk of bias in phylodynamic analyses. To reduce sampling bias, we took samples according to the number of estimated cases of the sublineage to improve the correlation between the number of genomes and the number of estimated cases by region per week. Correlations were estimated by Pearson coefficient. Three samples were taken for each sublineage with each samples being restricted to around 200 total genomes. Each sample was used to construct a maximum likelihood phylogeny using IQTREE2 with the GTR + F + I substitution model. The root-to-tip distance vs. sequence sampling time correlation was evaluated using Tempest, and likelihood mapping analyses were performed to determine the percentage of resolved quartets using 10000 quartets and the GTR + F + I substitution model.

### Bayesian phylogeographic analysis to analyse the migration patterns of sublineages between Peruvian regions

We used the Bayesian discrete phylogeographic model to estimate the time to the most recent common ancestor (MRCA) of each sublineage and the pattern of SARS-CoV-2 spread. We considered possible migrations between each Peruvian region and assumed reversible transition rates between locations. The analyses were conducted using the coalescent tree prior Bayesian integrated Coalescent Epoch PlotS (BICEPS), the General Time Reversible (GTR) substitution model with 4 gamma categories, and a strict clock with a uniform substitution rate prior distribution between 1.0E-4 and 0.01 substitutions per site per year (s/s/y). We assumed one partition during the analyses.

### Bayesian and MCMC runs

The analyses were performed using BEAST v2.6 software[63] with BEAGLE library[64] to speed up the run time. Four independent Markov Chain Monte Carlo were used with between 100–400 million iterations. Samples were diagnosed using Tracer v1.6[65] until they reached effective sample sizes of over 200 for all parameters. Maximum clade credibility trees (MCC) were summarized using the TreeAnotator

package[66]. To visualize and analyze trees and outputs, we used R with packages: ape, ggtree and treeio[67–70].

## Reporting summary

Further information on research design is available in the Nature Portfolio Reporting Summary linked to this article.

## Data availability

Publicly available datasets were analysed in this study. COVID-19 cases, deaths, stringency index and vaccinations at the country level were done based on the information of "Our World in Data" (https://github.com/owid/covid-19-data/tree/master/public/data). The number of cases by Peruvian regions were obtained from the "Plataforma Nacional de Datos Abiertos" available in https://www.datosabiertos.gob.pe/dataset/casos-positivos-por-covid-19-ministerio-de-salud-minsa. Genomic data was obtained from GISAID (https://www.gisaid.org). The accession codes of the sequences and associated metadata for epidemiology and for phylodynamic analyses used in this study are available in GISAID's EpiCoV database under EPI_SET_ID accession numbers EPI_SET_230526uh and EPI_SET_230526dk, respectively. All relevant output files are available following the instructions from the GitHub repository: https://github.com/sanjusare/Phylo_SARSCOV2_Peru.

## Code availability

Code is available on GitHub: https://github.com/sanjusare/Phylo_SARSCOV2_Peru. (https://doi.org/10.5281/zenodo.7976103)[71]

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

## Acknowledgements

We are very grateful to the GISAID Initiative and all its data contributors, i.e., the authors from the Originating laboratories responsible for obtaining the specimens and the Submitting laboratories where genetic sequence data were generated and shared via the GISAID Initiative, on which this research is based. We thanks to the Ricardo Palma University High-Performance Computational Cluster (URPHPC) managers Gustavo Adolfo Abarca Valdiviezo and Roxana Paola Mier Hermoza at the Ricardo Palma Informatic Department (OFICIC) for their contribution in programs and remote use configuration of URPHPC. Also, to Rodolfo Cardena Vigo, a manager of the High-Performance Computer from the Instituto de Investigaciones de la Amazonía Peruana, for his assistance in configurations and program installations. Funding for this work was provided by São Paulo Research Foundation (FAPESP) student fellowship 2015/13318-4 and 2022/00943-1 to S.J.A, Sociedade Beneficente Israelita Brasileira Hospital Albert Einstein (SBIBHAE) and Associação Brasileira de Gestão em Projetos (ABGP) for scholarship to S.J.A (SGPP: 3534-18), and Universidad Ricardo Palma. The funders had no role in study design, data collection and analysis, decision to publish, or preparation of the manuscript.

## Author contributions

Conceptualization: S.J.A., C.S.U.C., C.J.S., methodology: S.J.A., C.S.U.C., C.J.S, R.B., computing resources: M.Q.A., S.J.A., C.J.S., R.B., data curation: S.J.A., C.S.U.C., C.J.S., formal analysis: S.J.A., C.S.U.C., C.J.S., R.B., visualization: S.J.A., writing—original draft preparation: S.J.A., C.S.U.C., writing—review and editing: S.J.A., C.S.U.C., C.J.S., R.B., J.R.R.P., supervision: S.J.A., M.Q.A., R.B., J.R.R.P., funding acquisition: S.J.A., M.Q.A., J.R.R.P.. All authors read, provided critical review, and approved the final manuscript.

## Competing interests

The authors declare no competing interests.
