## [Peer Review File · Nature Communications]

Phylogenetic of SARS-CoV-2 during the second wave of COVID-19 in PeruEditorial Note: This manuscript has been previously reviewed at another journal that is not operating a transparent peer review scheme. This document only contains reviewer comments and rebuttal letters for versions considered at Nature Communications.

Reviewers' Comments:

Reviewer #2:

Remarks to the Author:

I revised an earlier version of this manuscript for an other journal. The authors have included responses to all my comments in their submission letter. The analyses and scope of the manuscript have changed substantially and I am satisfied that my concerns have been addressed. The current presentation of the phylogeographic analyses is very clear and indeed I wish more papers presented matrices with number of transitions, as in Figs 4 and 6.

However, I do think that the authors need to address two important points here:

1- In two separate sections, the authors state that a correlation coefficient of >0.5 of the root-to-tip distance as a function of sequence sampling time is sufficient for reliable molecular clock calibration. Surely this is an arbitrary cut-off. Indeed, there are no formal tests of temporal signal here, like date-randomisations or BETS, conducted here. Please do not use the correlation coefficient to determine whether molecular clock calibration is feasible. This statistic, and the root-to-tip regression in general, are meant as tools to inspect the data and detect any problems, like incorrectly dated samples, very divergent branches, and others. Now, I do not think that an actual test of temporal signal here is needed, because SARS-CoV-2 is generally amenable to molecular clock calibration, but at least mention the evolutionary rate estimate and check that it is similar to previous estimates.

2- The manuscript is plagued with typos and grammatical errors. I listed below a few examples of errors, but a careful read would certainly help pick up many other mistakes:

Please check the text carefully. E.g.

Line 76: "when a raise in the number" → the correct term is "rise"

Line 76: "again encourage the" → this is past tense, so "encouraged"

Line 108: "south America" → "South America"

Reviewer #3:

Remarks to the Author:

This manuscript has been thoroughly revised, and was a pleasure to read. I have only a few minor comments to address.

Introduction:

Line 27: 635 million is an outdated number by now (March 2023).

Line 155: Can you define quartets please?

Results/Discussion:

Line 145: Although I agree that one Lambda genome per each ~ 7400 Lambda cases will reduce sampling bias and is good practice, I think that it is important to recognise that this introduces another layer of bias (in that a variable percentage of true cases were being recorded in each

country).

Am I correct that you used 200 genomes from each Peruvian region for both the phylodynamic sublineage analysis and the phylogeographic analysis? Was this the same dataset? Did you use multiple replicates? It would be helpful if you could expand this methodology a little more, to ensure it can be replicated.

Reviewer #5:

Remarks to the Author:

This manuscript presents a fairly straight-forward analysis of Lambda and Gamma SARS-CoV-2 genome sequences that were sampled in Peru and nearby countries. The main purpose of the study is to test the hypothesis that Lambda originally emerged in Peru, where the earliest samples were reported, and to characterize the subsequent spread of this variant to other countries. The analytical methods employed by the investigators have become fairly standard for similar studies of regional SARS-CoV-2 epidemics - this is not a study about methodological innovation.

Following up on a comment from a previous reviewer, this version of the manuscript still requires some proof-reading to correct numerous grammatical errors and awkward phrasing. I have tried to provide some suggested corrections below.

Major comments:

* The GitHub repository links out to a Google Drive folder containing XML and Newick files corresponding to their analysis workflow outputs. Although it is important to provide access to study data to enable others to reproduce and verify the analysis, the XML files contain SARS-CoV-2 sequence data from GISAID. This redistribution of sequence data violates their Data Access Agreement.

* "we randomly take one Lambda genome per each ~7400 Lambda cases per week of each country" - does this down-sampling method exclude weeks where fewer than 7,400 cases were determined to be Lambda? Doesn't this mean that the earliest weeks (that would be the most informative about the origin of Lambda) were excluded from the down-sampled data?

* What are the mutations associated with the sublineages for Lambda and Gamma described in this study? I can understand relying on bootstrap support for having a reproducible partition of the tree into two or more subset trees (clusters). However, this partition is not necessarily the most informative for the purpose of characterizing the evolving distribution of infections among regions (e.g., Figure 6). This is similar to the modifiable areal unit problem.

* I am a bit concerned about the first Lambda genome that was reported to be sampled in Peru with a collection date of 2020-07-21 (hCoV-19/Peru/LIM-INS-8425/2020). The submission date associated with this genome in the GISAID database is 2021-11-05, or about a 15 month delay between sampling and publication. Is it possible that this is a misannotation? (See also lines 299-300.)

* I agree with a previous reviewer who noted that a better approach to evaluate the contribution of various factors to the epidemic spread is to map the migration rates associated with a discrete phylogeographic model to predictor variables (such as the distance between regions, or mean air traffic over a period of time) via a regression model. Otherwise the interpretation of results is limited.

The required data may not be readily available, however.

Minor comments

* The GitHub repository is clearly laid out and sufficiently documented that the reader should be able to reproduce their analysis. I would encourage the authors to follow some code style guidelines for their R scripts, such as using spaces around operators, breaking lines at a limit of 80 characters, and adding line breaks between logical sections of code.

* line 38-39, "In the first 90 days of the pandemic, the highest accumulated incidence rate [...]" - are the authors sure about Ecuador?

* I could not find details on the local polynomial regression method anywhere in the manuscript. Did the authors use LOESS? What were the smoothing parameters?

* lines 255-258: These numbers of transitions involving specific regions are difficult to interpret without providing the overall number of transitions.

* line 408, "between 1.0E-4 and 0.01", please provide units

* Figure 2, the legend should explain what the dashed vertical lines represent

* Figure 5, the first clade/sublineage in the Lambda tree seems to contain nearly all sequences except for a very small number of genomes. Is this subL1 or subL2? These labels should appear on the tree. In addition, subL2 is nested within subL1, so these do not appear to be monophyletic clades. This is fine, but the authors need to make it clear what they mean exactly for their clade definitions. The same is true for the Gamma clades.

* Figure S1 is really odd - why report the number of genomes per day? Why not per week? The sizing of this figure also makes it very difficult to see these counts. Are these sample collection dates, or submission dates?

* Figure S3 is unreadable.

* Figure S5, please clarify what you mean by "September sample" in the figure legend

Grammatical and typographical errors

* line 44, "[important] global prevalence" - awkward phrasing

* line 46, "(VOIs) [for] the World Health Organization" - replace with "by"

* line 52, "there has been a great effort to sequence genomes managing to sequence around 300 genomes per week" - this reads like the effort is the subject that is associated with the verb "managing"

* lines 76 and 78, "encourage[d]", "continue[d]"

* line 80, "a new [raise] of cases", replace with "rise"?

* line 86, "had a lethality rate of", should this be "case fatality rate"?

* line 87, "hits" should be past tense

* lines 100-101, "Alpha and Mu were not [such] successful in Peru" - replace with "so" or "as"?

* lines 119-120, the double-negative makes this unnecessarily difficult to follow. I suggest rewriting this as two criteria for identifying countries as potential origins for Lambda: (1) Lambda must have been sampled in at least two cities before April 2021, and (2) at least 15% of sequenced cases

sampled before April 2021 must have been classified as Lambda.

* line 128, "[impedes us] to include them in the analysis" - awkward phrasing

* lines 152-153, "presented more than 0.55 correlation coefficient that it is sufficient to get good calibration of the molecular clock" - awkward phrasing (also, which correlation coefficient?)

* line 234, "[impeded] that the correlations reach even higher R values" - awkward phrasing, and do you mean "implied"?

* lines 267-268, awkward phrasing

* line 270, "As previously mentioned, SubG2 could [have] emerged in the center [...]"

* lines 290-291, "All of them circulating in all the regions of Peru." This is not a complete sentence.

* lines 418-419, I believe these should be 70%, not "70" (a similar error was previously noted by a reviewer)

* line 438, "between [e]ach Peruvian region"

We thank the Editor and both reviewers for the time taken to read our manuscript and for the valuable comments that were provided.

Our responses are in blue font below the reviewer comments.

REVIEWER COMMENTS

Reviewer #2 (Remarks to the Author):

I revised an earlier version of this manuscript for an other journal. The authors have included responses to all my comments in their submission letter. The analyses and scope of the manuscript have changed substantially and I am satisfied that my concerns have been addressed. The current presentation of the phylogeographic analyses is very clear and indeed I wish more papers presented matrices with number of transitions, as in Figs 4 and 6.

However, I do think that the authors need to address two important points here:

1- In two separate sections, the authors state that a correlation coefficient of >0.5 of the root-to-tip distance as a function of sequence sampling time is sufficient for reliable molecular clock calibration. Surely this is an arbitrary cut-off. Indeed, there are no formal tests of temporal signal here, like date-randomisations or BETS, conducted here. Please do not use the correlation coefficient to determine whether molecular clock calibration is feasible. This statistic, and the root-to-tip regression in general, are meant as tools to inspect the data and detect any problems, like incorrectly dated samples, very divergent branches, and others. Now, I do not think that an actual test of temporal signal here is needed, because SARS-CoV-2 is generally amenable to molecular clock calibration, but at least mention the evolutionary rate estimate and check that it is similar to previous estimates.

Answer: We agree with the reviewer. We have modified the text to clarify the concern of the reviewer (lines 153-154, 262-263, 461-462, 501). Also, we have included supplementary figures (Fig S7) showing the estimated substitution rate in each of the phylogeographic analyses.

2- The manuscript is plagued with typos and grammatical errors. I listed below a few examples of errors, but a careful read would certainly help pick up many other mistakes:

Please check the text carefully. E.g.

Line 76: “when a raise in the number” → the correct term is “rise”

Line 76: “again encourage the” → this is past tense, so “encouraged”

Line 108: “south America” → “South America”

Answer: Corrections suggested by the reviewer were done (Lines 75, 107). Additionally, we have throughout reviewed the text carefully.

Reviewer #3 (Remarks to the Author):

This manuscript has been thoroughly revised, and was a pleasure to read. I have only a few minor comments to address.

Introduction:

Line 27: 635 million is an outdated number by now (March 2023).

Answer: We agree with the reviewer. We have updated the number of confirmed cases worldwide (Line 27).

Line 155: Can you define quartets please?

Answer: We have defined resolved quartets in the text (lines 159-161) and included a reference where the complete description of likelihood mapping and quartets is presented (line 153, Strimmer and Von Haeseler. 1997).

Results/Discussion:

Line 145: Although I agree that one Lambda genome per each ~7400 Lambda cases will reduce sampling bias and is good practice, I think that it is important to recognise that this introduces another layer of bias (in that a variable percentage of true cases were being recorded in each country).

Answer: We completely agree with the reviewer. We have mentioned this in the results (lines 150-151) and in the discussion sections (lines 365-367).

Am I correct that you used 200 genomes from each Peruvian region for both the phylodynamic sublineage analysis and the phylogeographic analysis? Was this the same dataset? Did you use multiple replicates? It would be helpful if you could expand this methodology a little more, to ensure it can be replicated.

Answer: Identification of sublineages and estimation of relative prevalences by sublineages were calculated with all the high-quality available genomes from Peru. On the other hand, phylogeographic analyses were done using a sample of genomes considering the number of cases by week/region. Triplicate samples were taken and we now included supplementary figures (Fig. S15 and S16) showing the results of the analyses by sample. Also, we have expanded the description of the methodology related to this (lines 480-482, 496-499).

Reviewer #5 (Remarks to the Author):

This manuscript presents a fairly straight-forward analysis of Lambda and Gamma SARS-CoV-2 genome sequences that were sampled in Peru and nearby countries. The main purpose of the study is to test the hypothesis that Lambda originally emerged in Peru, where the earliest samples were reported, and to characterize the subsequent spread of this variant to other countries. The analytical methods employed by the investigators have become fairly standard for similar studies of regional SARS-CoV-2 epidemics - this is not a study about methodological innovation.

Following up on a comment from a previous reviewer, this version of the manuscript still requires some proof-reading to correct numerous grammatical errors and awkward phrasing. I have tried to provide some suggested corrections below.

Major comments:

* The GitHub repository links out to a Google Drive folder containing XML and Newick files corresponding to their analysis workflow outputs. Although it is important to provide access to study data to enable others to reproduce and verify the analysis, the XML files contain SARS-CoV-2 sequence data from GISAID. This redistribution of sequence data violates their Data Access Agreement.

Answer: We have removed all the sequence data information from the .xml files. Newick and Nexus files were removed and just .log files of the runs were maintained for analysis verification.

* "we randomly take one Lambda genome per each ~7400 Lambda cases per week of each country" - does this down-sampling method exclude weeks where fewer than 7,400 cases were determined to be Lambda? Doesn't this mean that the earliest weeks (that would be the most informative about the origin of Lambda) were excluded from the down-sampled data?

Answer: We agree with the reviewer that the absence of genomes from the beginning of Lambda waves could affect the estimation of the origin of the Lambda variant. To analyze the effect of the inclusion of Lambda genomes available from weeks before Lambda reached 7400 estimated cases, we have taken a new set of subsamples following the same methodology previously described but also adding one randomly taken genome from those weeks (lines 457-460). The introduction of these genomes clearly will break the correlation between the number of genomes and the number of cases during these first weeks. Despite this, in this particular case, the conclusion that the Lambda variant more likely emerged in Peru did not change. Although it is important to highlight that one sample gave similar probabilities to the three countries. We have added the description of these analyses in the main text (lines 175-186)

* What are the mutations associated with the sublineages for Lambda and Gamma described in this study? I can understand relying on bootstrap support for having a reproducible partition of the tree into two or more subset trees (clusters). However, this partition is not necessarily the most informative for the purpose of characterizing the evolving distribution of infections among regions (e.g., Figure 6). This is similar to the modifiable areal unit problem.

Answer: We have identified the mutations that characterize each of the sublineages described in the analysis. The presence of a C or T in position 28849 is the main difference between SubL1 and SubL2 (C and T in SubL1 and SubL2, respectively) (Fig S11 and S12). However, because some groups in SubL1 also have a T in position 28849, we decided to perform phylodynamic analysis of Lambda with SubL1 and SubL2 together (SubL1+L2). On the other hand, SubG1, SubG2 and SubG3 are clearly defined by 6 positions: 3049,10116,22298,23599,23604,25613 (Fig. S12). We have add the description of this analysis in the results section (lines 217-218, 235-236, 241-242, 246-248, 265-266) and in the methodology section (lines 488-489).

* I am a bit concerned about the first Lambda genome that was reported to be sampled in Peru with a collection date of 2020-07-21 (hCoV-19/Peru/LIM-INS-8425/2020). The submission date associated with this genome in the GISAID database is 2021-11-05, or about a 15 month delay between sampling and publication. Is it possible that this is a misannotation? (See also lines 299-300.)

Answer: We agree with the reviewer but because we cannot confirm a misspelling we have decided to add a comment in the text mentioning the possibility of an incorrect annotation of the date of collection for this genome (lines 348-351).

* I agree with a previous reviewer who noted that a better approach to evaluate the contribution of various factors to the epidemic spread is to map the migration rates associated with a discrete phylogeographic model to predictor variables (such as the distance between regions, or mean air traffic over a period of time) via a regression model. Otherwise the interpretation of results is limited. The required data may not be readily available, however.

Answer: We agree with the reviewer in this respect. Unfortunately, the data required to do this is not readily available. We add a comment about this in the discussion section (lines 375-378).

Minor comments

* The GitHub repository is clearly laid out and sufficiently documented that the reader should be able to reproduce their analysis. I would encourage the authors to follow some code style guidelines for their R scripts, such as using spaces around operators, breaking lines at a limit of 80 characters, and adding line breaks between logical sections of code.

Answer: We have added line breaks between logical sections of the code with comments explaining the objective of each section. We also broke lines at a limit of 80 characters when possible and implemented space around operators.

* line 38-39, "In the first 90 days of the pandemic, the highest accumulated incidence rate [...]" - are the authors sure about Ecuador?

Answer: Yes, based on the analysis performed by Acosta, 2020. (lines 37-39).

* I could not find details on the local polynomial regression method anywhere in the manuscript. Did the authors use LOESS? What were the smoothing parameters?

Answer: We have included the smoothing parameters in the methodology section (lines 434-436).

* lines 255-258: These numbers of transitions involving specific regions are difficult to interpret without providing the overall number of transitions.

Answer: We have included overall number of transitions in the results section (lines 289-290, 297-298, 308-309, 318-319) and in the Figure 6.

* line 408, "between 1.0E-4 and 0.01", please provide units

Answer: We have specified the units (lines 474-475, 514).

* Figure 2, the legend should explain what the dashed vertical lines represent

Answer: We included an explanation of dashed vertical lines in the legend of this figure.

* Figure 5, the first clade/sublineage in the Lambda tree seems to contain nearly all sequences except for a very small number of genomes. Is this subL1 or subL2? These labels should appear on the tree. In addition, subL2 is nested within subL1, so these do not appear to be monophyletic clades. This is fine, but the authors need to make it clear what they mean exactly for their clade definitions. The same is true for the Gamma clades.

Answer: We added labels with the assigned sublineage name in their parental nodes. (Figure 5).

* Figure S1 is really odd - why report the number of genomes per day? Why not per week? The sizing of this figure also makes it very difficult to see these counts. Are these sample collection dates, or submission dates?

Answer: We have modified this figure to improve its clarity. In the legend of the figure we specify that we are referring to collection weeks (Figure S1).

* Figure S3 is unreadable.

Answer: We have improved the presentation and resolution of this figure.

* Figure S5, please clarify what you mean by "September sample" in the figure legend

Answer: We have clarified the figure and the legend of this figure.

Grammatical and typographical errors

* line 44, "[important] global prevalence" - awkward phrasing

* line 46, "(VOIs) [for] the World Health Organization" - replace with "by"

* line 52, "there has been a great effort to sequence genomes managing to sequence around 300 genomes per week" - this reads like the effort is the subject that is associated with the verb "managing"

* lines 76 and 78, "encourage[d]", "continue[d]"

* line 80, "a new [raise] of cases", replace with "rise"?

* line 86, "had a lethality rate of", should this be "case fatality rate"?

* line 87, "hits" should be past tense

* lines 100-101, "Alpha and Mu were not [such] successful in Peru" - replace with "so" or "as"?

* lines 119-120, the double-negative makes this unnecessarily difficult to follow. I suggest rewriting this as two criteria for identifying countries as potential origins for Lambda: (1) Lambda must have been sampled in at least two cities before April 2021, and (2) at least 15% of sequenced cases sampled before April 2021 must have been classified as Lambda.

* line 128, "[impedes us] to include them in the analysis" - awkward phrasing

* lines 152-153, "presented more than 0.55 correlation coefficient that it is sufficient to get good calibration of the molecular clock" - awkward phrasing (also, which correlation coefficient?)

* line 234, "[impeded] that the correlations reach even higher R values" - awkward phrasing, and do you mean "implied"?

* lines 267-268, awkward phrasing

* line 270, "As previously mentioned, SubG2 could [have] emerged in the center [...]"

* lines 290-291, "All of them circulating in all the regions of Peru." This is not a complete sentence.

* lines 418-419, I believe these should be 70%, not "70" (a similar error was previously

noted by a reviewer)

* line 438, "between [e]ach Peruvian region"

Answer: Corrections suggested by the reviewer were implemented. Additionally, we have throughout reviewed the text carefully.

Reviewers' Comments:

Reviewer #2:

Remarks to the Author:

The authors have made substantial revisions to their manuscript and I am satisfied that all my concerns have been addressed. Thus, I am happy to recommend this article for publication.

Reviewer #3:

Remarks to the Author:

I am happy with the edits the authors have made in response to my comments.

My only final recommendation would be to make the trees in Fig 4ab to be made bigger/better resolution.

Reviewer #5:

Remarks to the Author:

I thank the authors for addressing the issues that I raised in my previous review. I only have some minor comments for the revised version that the authors may want to resolve at their discretion.

- Abstract, lines 21-22: "Finally, phylodynamics by Peruvian regions showed that [...]" awkward phrasing
- line 25: "since the first cases of [unexplained viral] pneumonia" (?)
- line 31: "The first wave range[d]"
- lines 33-34: "causing the highest number[s] of reported cases"
- lines 59-60: "and together the date of emergence has to be revisited." awkward phrasing
- lines 64-65: "Bayesian phylogenetics to determine the most likely date" - Bayesian inference is used to estimate a posterior probability distribution over the space of model parameters - not a point estimate.
- line 86: "But [this time], just a modest"
- line 89: "In contrast, the lethality rate decreased" Do you mean case fatality rate?
- line 116: "Based on [evidence] that Lambda has a comparable"
- line 126: "Thus, we cannot [rule out] these countries and cities [...]"
- line 128: "[...] because there are no available genomes from them." This seems redundant given previous statements in this paragraph.
- lines 144-145: "[...] we aim to determine which of them has more probability to be the origin." awkward phrasing

- line 153: "Then, we analyzed the phylogenetic content by likelihood mapping [...]" Unclear what is meant here.

- line 168: Has "HPD" been defined? (highest posterior density)

- line 169-170: "indicating that Lambda ha[d] already been circulating"

We thank the Editor and reviewers for the time taken to read our manuscript and for the valuable comments that were provided.

Our responses are in blue font below the reviewer comments.

REVIEWER COMMENTS

Reviewer #2 (Remarks to the Author):

The authors have made substantial revisions to their manuscript and I am satisfied that all my concerns have been addressed. Thus, I am happy to recommend this article for publication.

It was our pleasure to satisfy the concerns of the reviewer

Reviewer #3 (Remarks to the Author):

I am happy with the edits the authors have made in response to my comments.

It was our pleasure to satisfy the concerns of the reviewer

My only final recommendation would be to make the trees in Fig 4ab to be made bigger/better resolution.

Figure 4 and, in general, all the figures will be sent at the highest resolution for final publication of the manuscript.

Reviewer #5 (Remarks to the Author):

I thank the authors for addressing the issues that I raised in my previous review. I only have some minor comments for the revised version that the authors may want to resolve at their discretion.

It was our pleasure to satisfy the main concerns of the reviewer

- Abstract, lines 21-22: "Finally, phylodynamics by Peruvian regions showed that [...]" awkward phrasing
- line 25: "since the first cases of [unexplained viral] pneumonia" (?)
- line 31: "The first wave range[d]"
- lines 33-34: "causing the highest number[s] of reported cases"
- lines 59-60: "and together the date of emergence has to be revisited." awkward phrasing
- lines 64-65: "Bayesian phylogenetics to determine the most likely date" - Bayesian inference is used to estimate a posterior probability distribution over the space of model parameters - not a point estimate.
- line 86: "But [this time], just a modest"
- line 89: "In contrast, the lethality rate decreased" Do you mean case fatality rate?
- line 116: "Based on [evidence] that Lambda has a comparable"
- line 126: "Thus, we cannot [rule out] these countries and cities [...]"

- line 128: "[...] because there are no available genomes from them." This seems redundant given previous statements in this paragraph.
- lines 144-145: "[...] we aim to determine which of them has more probability to be the origin." awkward phrasing
- line 153: "Then, we analyzed the phylogenetic content by likelihood mapping [...]" Unclear what is meant here.
- line 168: Has "HPD" been defined? (highest posterior density)
- line 169-170: "indicating that Lambda ha[d] already been circulating"

All the suggestions made by the reviewer has been implemented. In the case of the suggestion of line 153, we have modified throughout the text the term "phylogenetic content" to avoid confusions. We directly mentioned "resolved quartets" that has a precise definition in lines 153-155).